## Citizen Science

ecological value; ecosystem capital; land-water-biomass resources; proof-of-concept; societal empowerment.

**Author for correspondence:**
J. Argüello
E-mail: jazmin.arguello@ens-lyon.fr

# Ecosystem natural capital accounting: The landscape approach at a territorial watershed scale

Jazmin Argüello[1] , Jean-Louis Weber[1,2] and Ioan Negrutiu[1]

[1]Institut des Systèmes Complexes (IXXI) and Laboratoire de Reproduction et Développement des Plantes, Université de Lyon, UCB Lyon 1, École Normale Supérieure de Lyon, INRAE, CNRS, 46 Allée d'Italie, 69364 Lyon Cedex 07, France; [2]European Environment Agency, Scientific committee, Frankrigshusene 9, 1 tv, 2300 Copenhagen S, Denmark

## Abstract

Most approaches to estimate ecological value use monetary valuation. Here, we propose a different framework accounting ecological value in biophysical terms. More specifically, we are implementing the ecosystem natural capital accounting framework as an operational adaptation and extension of the UN System of Economic and Environmental Accounting/Ecosystem Accounting. The proof-of-concept study was carried out at the Rhône river watershed scale (France). Four core accounts evaluate land use, water and river condition, bio-carbon content of various stocks of biomass and its uses, and the state of ecosystem infrastructure. Integration of the various indicators allows measuring ecosystems overall capability and their degradation. The 12-year results are based on spatial–temporal geographic information and local statistics. Increasing levels of intensity of use are registered over time, that is, the extraction of resources surpasses renewal. We find that agriculture and land artificialisation are the main drivers of natural capital degradation.

## 1. Introduction

Anthropogenic changes to ecosystem functions and derived services are essentially due to over-consumption of primary resources, human population increase and technology-driven ecosystem use intensification (Carpenter et al., 2009; Millennium Ecosystem Assessment, 2005). The dominant economic thinking drives the development discourse (Norgaard, 2010; Rees, 2015a), and governments or national, regional and local companies do not keep systematic natural capital accounting records.

The depletion resulting from this consumption of renewable assets leads to ecosystem degradation and to the loss of their ability to provide goods and services. This is equivalent to generating negative externalities due to the consumption of ecological capital (an equivalent to depreciation or unpaid costs). Thus, the situation represents economic and political risks. Such risks are matters of national security and sovereignty, and they burden present and future generations. The Swiss Re Institute (2020) estimates that about half of the global gross domestic product (GDP) depends on high-functioning ecosystems (see also Dasgupta et al., 2021) and Chaplin-Kramer et al. (2019) show that people's needs for nature and the ability of nature to provide them increasingly diverge.

In society-at-large, these notions are gaining ground (Convention on Biological Diversity, 2020; Dasgupta et al., 2021; Sustainable Development Goals, 2020; World Wildlife Fund, 2019) but remain far from being translated into coherent and convergent actionable levers of change. One such lever, namely the development of strong sustainability (Table 1) building on the evaluation of ecosystem services has been slow to gain ground.

Environmental evaluations, essential tools to socio-ecological system approaches (Bourgeron et al., 2018; Li et al., 2020; World Wildlife Fund, 2019), are being developed to assess the direct or indirect impacts of externalities on ecological systems and their productivity. These

developments have contributed to the emergence of a broad range of approaches, methodologies and environmental evaluation instruments (Mazza et al., 2013; Organisation for Economic Co-operation and Development, 2012; United Nations Environment Programme, 2014; Weber, 2014; West, 2015), aimed to integrate the environment and natural resources into economics-based national accounting frameworks (Caldecott et al., 2013; Weber, 2018). Despite such accomplishments, a recent survey concluded that 'there is very little use of natural capital accounts for public policy decisions and, more so, in developing countries' (Recuero Virto et al., 2018). A significant reason is that political and economic decision-making and societal choices are restricted by the following limitations or contradictions in the current instruments (Argüello et al., 2020):

1. Ecosystems are often reduced to their monetary value and are aggregating distinct categories of ecosystem capital, thus hindering other possible frameworks.
2. Methodologies often target the 'intensity of resource use' and measure flow values rather than changes in stocks.
3. Methodologies attempting to encapsulate the GDP within more or less strict ecological limits and sustainability are heterogeneous.
4. Ecosystem service assessments are confronted with the challenge of the interconnected and multifunctional nature of services (e.g., avoiding double-counting or incomplete services counting or none at all).

Here, we implement a novel approach, called ecosystem natural capital accounting (ENCA), which instead considers accounts in biophysical terms.

ENCA was designed to address the notions of ecological value and ecosystem potential (Table 1) and measure degradation. The publication in 2014 of the ENCA Quick Start Package (Weber, 2014) by the CBD Secretariat intended to support with operational methodologies the implementation by countries of the System of Economic and Environmental Accounting (SEEA) experimental Ecosystem Accounting. Considering accounts in biophysical terms, ENCA is broadly compatible with the UN Economic and Environmental Accounting System volume on Ecosystem Accounts adopted by the UN Statistical Commission in 2021 (United Nations—System of Environmental Economic Accounting, 2021). However, regarding monetary assessments, the SEEA-EA cornerstone is the valuation of the benefits provided by ecosystem services and assets, while ENCA approach to biophysical degradation leads to the calculation of unpaid restoration costs to meet the injunction of no net degradation of ecosystems. In other words, to estimate an 'ecological value' we depart from existing monetary valuation, which indirectly legitimise a right to exploit ecosystems, to biophysical valuation, which instead tends to consider the degradation and thus the associated biophysical debt.

The ENCA tool is based on four core accounts: land cover, water and rivers, bio-carbon and the functional services provided by the ecosystem infrastructure. This corresponds to an extension of carbon budgets (Intergovernmental Panel on Climate Change, 2006) with additional geo-physicochemical and biological parameters.

The purpose of ENCA (Figure 1) is to

1. describe how resource stocks and flows change over time to determine trends reflecting the real availability of each resource for use,

2. characterise and quantify ecosystem states with common ecological metrics to ultimately measure depletion, degradation or improvement, considering intensity of resource use in quantitative terms and diagnose ecosystem health.

Ecosystem degradation is considered as the loss of ecosystem assets' ecological value. As there is no metric for ecological value, ENCA proposes a new framework and tool to compute a unit equivalent for measuring the ecological value, that is, an aggregate summarising the various quantitative and qualitative changes recorded in the accounts, a currency of the ecosystem capital health condition. In other words, ENCA evaluates how natural assets are interconnected to respond to various pressures and reveals degradations relating to the use of the ecosystem.

The present work implements ENCA at the watershed scale and landscape resolution (Table 1). The evaluations at the river watershed scale are pertinent politically, economically, socially and ecologically because they offer systemic spatial coherence, a critical factor in managing territorial resources and ecosystem services.

The following sections report on the ENCA proof-of-concept tool applied at the Rhône river watershed scale (France) to describe the main steps of the ENCA methodology, present the results in the form of accounting tables and in cartographic form to facilitate their visualisation and analysis, supplement the conventional accounts used by private and public organisations with information allowing them to integrate in their reporting systems the accountability to ecosystem use and discuss the importance of its deployment as an aid to decision-making and societal empowerment.

## 2. Materials and methods

### 2.1 The Rhône river watershed

The report evaluates the ecosystem capital of the Rhône river watershed (Supplementary Material S1) during 2000–2012, dates corresponding to the European CORINE Land Cover Maps available at the time of the study. CORINE Land Cover (CLC) provides medium resolution maps (circa 1/100 000) updated every 6 years for the 39 European Environment Agency member and partner States (CORINE, 2017). The 2018 CLC was delivered too late for being used in this study.

The territory is a transboundary basin shared by France and Switzerland. The 97,800 km$^2$ area encompasses several valleys and rivers in three major regions in Europe, alpine, continental and Mediterranean (Olivier et al., 2009). This includes five administrative regions and seven departments in the French part (more than 90% of the basin), and three cantons (Vaud, Valais and Geneva) in the Swiss portion.

We have focused our research on the French part of the watershed because the data management systems between the two countries are not fully compatible.

### 2.2 The ENCA methodological frame

There are four core accounts that evaluate

1. Land cover and use,
2. Water quantity and quality accounts,
3. Bio-carbon accounting and
4. Ecosystem landscape and rivers infrastructure.

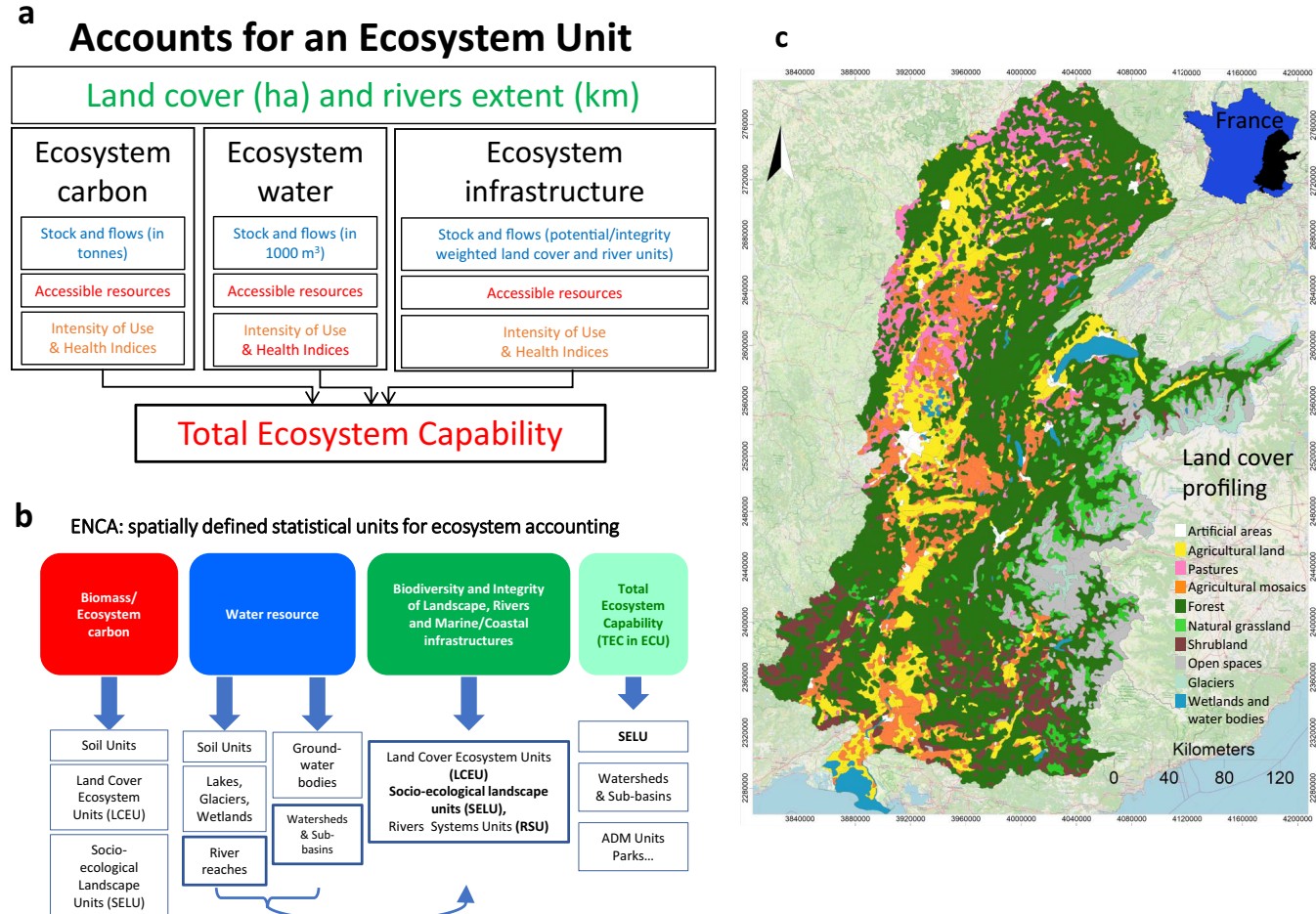

**Fig. 1.** The ENCA accounting framework and spatially defined statistical accounting units. (a) Articulation and integration of ENCA components. The land cover and river extent account structures the three interconnected and integrated thematic accounts (bio-carbon, water and ecosystems infrastructure), while defining and implementing the statistical units for accounting. River systems are considered a particular land-cover type measured in length and run-off instead of area. Combining landscape and river systems makes it possible to blend their assessment into the infrastructure integrity account (see diagram (b) below). Thematic accounts are made using statistical and geographic data of land and river ecosystems. The protocol combines quantitative and qualitative variables. A structure common to three accounts: the landscape-scale spatial unit, the socio-ecological landscape unit (SELU) (see below)—provides internal integration. Quantitative accounts record stocks and flows for measuring the resource accessible (without depletion) and compare it to the total use. They deliver an index of intensity of use. For each account, qualitative elements help diagnose ecosystem health, as summarised in an index of landscape potential (land and river). The indices of intensity of use and health are combined to measure the mean internal ecological unit value for each component expressed in ecological capability units (ECU). Being expressed in the same unit, the resources of each thematic account can be aggregated to calculate the headline indicator of ENCA: the value of ecological capital, expressed in total ecosystem capability. The colours in the diagram indicate the degree of robustness of data sources and derived indicators. Green, very good; blue, good; orange, average; red, poor. (b) Diagram summarizing the different spatial meshes used to establish the accounts (ENCATs, UZHYD, SELU, administrative units) and illustrates the diversity of the observation levels and the corresponding sources of data mobilised during this work. The landscape-scale spatial unit, the SELU is common to the bio-carbon, water and ecosystems infrastructure accounts and provides internal integration. SELUs are statistical-geographic ecosystem units defined by a combination of dominant land-cover types (DLCT; see the map in (c) below and Supplementary Material S4a) and their respective physical properties, such as water circulation within river sub-catchments or a class of altitude. ECU values are compiled by SELU, the landscape systems where components combine and where basic ecosystem/land-cover units (agricultural land, pasture, forest, lakes and so forth) exchange and interact. (c) Map of the SELUs of the Rhône river watershed defined by the intersection of the 10 DLCT and catchments (e.g., watershed limits from BD Carthage, SANDRE, and OFEV sources). The 10 DLCTs are generated by the aggregation of 44 CLC classes. Note that natural grasslands are exclusively present in sub-alpine areas.

The latter three represent thematic accounts framed in the land accounts and articulated by socio-ecological landscape units (SELUs) (Figure 1).

For each thematic account, the data fall into four sets of accounting tables (available at http://www.ecosystemaccounting.net/?page_id=173). A sample of a workflow is shown in Supplementary Material S2a and of an accounting table in Supplementary Material S2b.

The detailed methods can be found at https://osf.io/j93xu/ (ENCA. Proof-of-concept folder). Table 2 presents the definitions and calculation formulas of main indicators. The results are based on spatial–temporal geographic information and socio-ecological units.

### 2.3 Land cover and river extent mapping, and conception of statistical units and account indicators

While in national accounts the statistic units are essentially legal in nature, the accounting units in ENCA are essentially geographical-spatial areas, where information is collected,

**Table 1.** Glossary

| Notion and definition | Observations/Interdependencies |
|---|---|
| *Critical areas or hotspots of ecological degradation.* Areas showing highest rates of ecosystem capital degradation (stocks, resource accessibility and ecosystem health) or highest intensity of use indices. | |
| *Ecological value:* non-monetary assessment of ecosystem integrity or health through indicators determining critical thresholds and minimum requirements for ecosystem services provision. *Economic valuation:* the process of expressing a value for a particular good or service in a certain context (e.g., of decision-making) in monetary terms. The monetary value results from market transactions. Both market and ecological values are social constructs, the former measuring utilities, the latter assessing intrinsic ecosystem dimensions. Ecosystem assets and services that lack market have no price. However, they have value that the economic values cannot replace (Gómez-Baggethun & Barton, 2013; Walker, 2005; Weber, 2018). The value of the ecosystem capital is considered an alternative to the use value, the latter being directed by utility and profit maximisation. The use value implies that the natural capital consists of stocks of monetary assets, the capital degradation being defined as loss of monetary assets value. The ecological value in ENCA integrates physical stocks in socio-ecological landscape units (SELU; see also Figure 1b,c). | The ecological value differs from economic value as it does not consider the monetary benefits provided by the ecosystem services but the state, that is, the health, of ecosystems functions (Rapport et al., 1998). Therefore, the ecological value restricts possibilities of substitution to the ecosystem capital itself (strong sustainability), while targeting the long-term functions of the ecosystem measured through their health, potential or capability to supply ecosystem services (Weber, 2014). The change of paradigm from monetary valuation to ecological evaluation is fundamental in a global change context and the necessity to embed the economy in nature. Using it would allow: –making the economy and other human activities compatible with the regenerative and assimilative capacities of the biosphere (Rees, 2015b; Røpke, 2004;) –targeting the social value of ecosystem capital instead of use value (Ekins et al., 2003; Gómez-Baggethun & Barton, 2013) –analysis to envisage the amortisation of an ecosystem's capital degradation through accounting norms and standards required for efficient ecosystem management and conservation actions. |
| *Ecosystem health* implies maintaining the system's organisation, functions and autonomy over time (Rapport et al., 1998). The integrity of Earth ecosystems is evaluated in terms of productivity, morphological and functional diversity and resilience to stress. Ecosystem health in this work refers to ecosystem condition or state, a constituent of the bidirectionally coupled socio-ecological framework (Binder et al., 2013). *Ecosystem potential* is the capability of a given ecosystem category to maintain its functions and provide the range of services that they supply. It is estimated in terms of health status and the intrinsic injunction of no net degradation of ecosystems. It is deliberately associated with an ecosystem response capacity. In ENCA, the potential is expressed in ecological capability units (ECU) for each ecosystem category. | Capability and potential are synonymous, but the latter term is frequently used in the sense of absolute potential referring to the natural condition (e.g., climax or a pre-industrial reference) while capability refers to a social target (e.g., no net degradation in reference to a given year, such as 1990, the baseline year of the United Nations Framework Convention on Climate Change). The potential approach has some similarities with the human appropriation of net primary production concept (Haberl et al., 2007). |
| *Landscape* (the) is a representation of sociocultural dynamics at an organisation level that can integrate structural, functional and compositional processes of biodiversity (Noss, 1990). | The landscape's importance has been highlighted mainly in land-use decisions. Such decisions are strongly influenced by local socio-political priorities that change through time (Mace, 2013; Zvoleff & An, 2014) and increasingly mobilise participatory approaches. The modelling at various scales of spatio-temporal trends in ecosystem services, supply and demand is being used in landscape management, spatial planning and regional development and financial policies (Boesing et al., 2020; Rieb & Bennett, 2020). |
| *Sustainability* A characteristic or state whereby the needs of the present and local population can be met without compromising the ability of future generations or population in other locations to meet their needs (Millennium Ecosystem Assessment, 2005; The Economics of Ecosystems and Biodiversity, 2008) Economic benefits result from combining several types of capital: produced, human, social, cultural and natural. In this perspective, aggregates of total or inclusive wealth cover possible substitutions between these types of capital, as acknowledged with the concept of weak sustainability. Its foundation is the substitutability among various forms of capital—gains in other capital forms can offset the loss of natural capital. The bottom line of strong sustainability is that substitutions are possible within but not between capital categories. Ecosystem capacities must be maintained so there will be no net degradation. | Strong sustainability, considered here in terms of an inclusive social and ecological system (Binder et al., 2013; Downing et al., 2020; Fischer-Kowalski & Steinberger, 2017), constitutes a frame allowing to address the ecosystem capital of a given territory through the lens of ecological rather than monetary value. In the search of a single indicator of strong environmental sustainability, Ekins et al. (2019) propose a dashboard of environmental sustainability indicators across a range of environmental and resource issues (Source, Sink, Life-Support and Human Health and Welfare). Data availability remains the major limitation towards the computation of lead indicators that form the thematic overview. |
| *Watershed* Also named water catchments or river basins, watersheds are important conceptual frameworks and natural systems for investigating complex socio-ecological processes (Jenkins et al., 2018; Parkes et al., 2010). They are functionally coherent hierarchical networks that can mobilise social and territorial actors and institutions located within their boundaries through a shared history of social and environmental issues (such as land-use policies, water governance and ecosystem management) and local knowledge and know-how. The purpose is 'building better and more resilient connections between institutions and ecological resources. All too often, administrative boundaries divide vital ecological resources, which make nonsense of the natural landscape. This is especially the case with rivers and wider watersheds, where the geographic integrity of the river basin is rarely matched by an administrative system with the powers required to manage upstream-downstream interactions' (Toulmin, 2017). | A political resource space where decisions are made on territorial resources in the form of governed projects that reflect the potential and the specifics of the corresponding resource space. That context facilitates coordinating disjoint public policies (water, land, agriculture, health, environment) and private sector activities with a long view on the common purpose (public interest). |

Abbreviations: ECU, ecological capability units; ENCA, ecosystem natural capital accounting.

**Table 2.** Definitions, calculation formulas, explanations on thematic indicators and considerations on data analysis

| Account | Indicators | Definitions and calculations | Observations/details |
|---|---|---|---|
| *Water* | Stocks | Calculated as the sum of the<br>–mean standard river-kilometre (SRKM or river unit = discharge by length, System of Environmental-Economic Accounts for Water, 2009),<br>–lakes and reservoirs (Carthage 'Hydrologie surfacique' for areas, Wikipedia for volumes of large lakes, default values for others),<br>–ground water (renewable water estimated from deep percolation, draining to rivers and abstraction of groundwater),<br>–snow and glaciers (Corine land cover for areas 2000, 2006, and 2012 and measurement of depth and change on a few points),<br>–soil water (ESDAC database of soil depth (down to 1 m) and stones content combined with soil humidity).<br>And the balance of precipitation flows (From sbwEWA data resampled against WorldClim), total actual evapotranspiration (TAE; from sbwEWA resampled against WorldClim, overlaid with CLC Agriculture classes, minus TAE induced by irrigation), and Abstraction (database on water abstraction of the RMC water agency). | Meteorological data come from a dataset of annual data 2000–2012, assimilated for the purpose of the swbEWA modelling exercise. It was kindly provided to the project by Kurnik et al. (2014). The coarse spatial resolution is compensated by the consistency of the variables on rainfall, actual evapotranspiration, soil humidity and runoff, important for deriving accounting results. |
| | Accessibility | The accessible resource is the surplus, not all water stock can be exploited as a resource.<br>For lakes and reservoirs, estimated by default as 10% of stock.<br>For rivers, the Total inflows—Reserved runoff (runoff exceeding the 3rd quartile of the period—2.5% of mean annual discharge). | Accessible resource is the amount of a resource that is accessible for uses in a sustainable way. It is not the stock itself nor the total stock plus inflow. Accessible resource is calculated by adjustment of the 'available resource' from all the elements which limit its use: respect of sustainable yields to avoid depletion, timeliness, distance, affordable economic costs of operation, respect of environmental norms and other legal constraints. |
| | Use | Estimated from RMC water agency database on water abstraction. | Spatial distribution of use by sector. |
| | Intensity of use | Corresponds to accessible water resources/use, if accessible water resources > use, then the index is 1. | |
| | health index | This assessment was done using indicators of 'good' or 'bad' water quality and vulnerability to nitrates according to the French water framework directive. The dependence on artificial inputs was evaluated as the agriculture dependence to irrigation measured by the ratio of green water (actual evapotranspiration) to irrigation water. | The indicators were scored to obtain an index where below one means 'poor' and one "good" health. |
| *Bio-carbon* | Stocks | Biomass is the sum of:<br>1. Bio-carbon tonnes of trees by hectare were calculated using the volume of trees and area (m3 ha-1) from the forest inventory at department level (IGN, 1992–2004), the vector database of forested polygons from the IGN (2006–2017), the tree relative cover in the watershed (Hansen et al., 2013), and conversion coefficients to biomass dry weight and to carbon content (Intergovernmental Panel on Climate Change, 2003; 2006),<br>2. Bio-carbon in litter and deadwood<br>The estimation of this stock corresponds only to dead-standing trees and wind-throw in forested areas. The carbon tonnes by hectare were calculated from the volume and area (m3 ha-1) of the forest inventory at regional level (IGN, 2011–2015), the spatial data of forested area from the IGN (2006–2017), and coefficients of the Intergovernmental Panel on Climate Change (2003; 2006), and<br>3. Bio-carbon in soil using data mining of soil samples with a theoretical sampling distance of 16 km, and measurements in situ from 2000 to 2009, and a cross-validation scheme (Mulder et al., 2016). The 90-m-resolution product from INRA (2018) was resampled to 1-ha pixel-size in SAGA-GIS. | The ecosystem carbon basic account describes the balance of stocks and flows and their relationships, in tonnes of carbon. |

**Table 2.** (Continued)

| Account | Indicators | Definitions and calculations | Observations/details |
|---|---|---|---|
| | Accessibility | Not all biomass can be exploited as a bio-carbon resource, only a surplus. All the net ecosystem production (NEP) was assumed to be accessible. NEP is the difference between net primary production (NPP, data from MODIS-NASA) and Heterotrophic respiration ($H_R$). $NPP = GPP - A_R$ where GPP means Gross Primary Production (data from MODIS-NASA) and $A_R$ autotrophic respiration. We assumed that heterotrophic respiration equals autotrophic respiration, thus $H_R = GPP - NPP$ | Accessible resource is defined above in the water account. |
| | Use | Total withdrawals of bio-carbon correspond to the sum of the extraction of carbon through: 1. Agriculture harvest crops. The geographic data of the crop plots extracted from the graphical land parcels (RPG, 2012). The statistics were downloaded from AGRESTE (2012). 2. Wood removals. The vector database of forested area (IGN,1992–2004) served for the spatialisation of the wood removal statistics (except energy) for 2000 (source Weber, 2018), 2006, and 2012 (AGRESTE, 2005–2017). 3. Withdrawals of animal bio-carbon. The cattle statistics downloaded from AGRESTE (2000–2015) and spatialised in the pasture class years 2000, 2006, and 2012 from CORINE (2017). 4. Loss of bio-carbon due to land-use change. The number of hectares and the type of change of land use were computed from the land cover CORINE (2017) dataset for changes from 2000 to 2006. The sum of the corresponding type stocks was multiplied by the number of hectares changed. This period was assumed as the median and used for the other two periods (namely 2006 and 2012). | |
| | Intensity of use | Net ecosystem production (corresponding to net ecosystem accessible carbon) divided by use. | The intensity of carbon use index is the ratio of Net accessible resource surplus to Total use of ecosystem bio-carbon. Values greater than one were set to one. |
| | health index | Indicators of ecosystem health regarding bio-carbon are changes such as the mean age of forest or fish stocks, and vulnerability to fire. An additional health indicator is the dependency of bio-carbon production on fossil energy inputs. | Data not available, therefore one was set by default. |
| *Ecosystem Infrastructure* | Accessibility and potential versus Services | Unlike carbon and water, where the accessible resource exists independent of any actual use, intangible ecosystem services need to be both accessible and actually physically accessed to exist. Therefore, they can be measured only indirectly. Because intangible ecosystem services are not additional, it is preferable to measure the potential of ecosystems to provide them. This potential is assessed from a system perspective considering their robustness and integrity. | Provisioning services (food, energy, timber and fibre, drinking water, among others) are tangibles incorporated by the economy (formal and informal) into commodities, duly recorded in ENCA in the bio-carbon and water accounts. Regulating services (maintaining the quality of air and soil, providing flood and disease control, pollinating crops…) and socio-cultural services are intangible and linked to places. |
| | NLEP | The macro indicator measures terrestrial ecosystem integrity (Weber et al., 2008), that integrates the land cover artificiality/naturalness (scores from 0 to 100 according to land-cover classes), the areas representing high species/habitats diversity (NATURILIS, scores from 0 to 10 were assigned according to international and French designations of protected areas, the max value was around 100 when all overlapped), and their connectivity (Moser et al., 2007). | Heterogeneities were mostly observed in the ecosystem infrastructure account for integrity and biodiversity parameters. NLEP—No time series available, except for land cover. |

**Table 2.** (Continued)

| Account | Indicators | Definitions and calculations | Observations/details |
|---|---|---|---|
| | NREP | The macro indicator measures river ecosystem integrity (Weber et al., 2008). This is calculated as the geometric mean of the river condition potential, also called river ecosystem background (SRKM_R x ecological status index, i.e., SRKM_R weighted by data on the ecological status of water streams/water bodies for 2009 and 2015 from Cartograph 'EauFrance, scores from 0.3 to 1), and the NATRIV (NATURILIS scores for rivers). | NREP—No time series available. The ecological status integrates chemical, biological, and morphological parameters. The ecological status of some rivers is not reported, missing values equal to one. |
| | TEIP | Sum of NLEP and NREP by SELU | |
| | health index | The biodiversity index summarises symptoms of distress. The evaluation was done using the Reporting under Article 17 of the Habitats Directive. The assessment of the conservation status of habitats and species of community interest is based on information on status and trends of species populations or habitats, and on information on main pressures and threats. | Values above one mean "good" health and values below one mean "poor" health. |
| *Account Integration (partial)* | SELU | Socio-ecological landscape unit. The landscape-scale spatial unit, a combination of dominant land cover types (DLCT) and catchments. | |
| | Unitary value (in ECU) | The geometric mean of intensity of use index and health index of each account. | |
| | Total ecosystem capability | The ECU values of thematic accounts are aggregated to amount the value of the ecological capital. | The ECU is the common metrics allowing the integration of accounts through aggregation in a composite indicator. |

*Note.* Based on available data for each core account, the quantitative stocks and use balances, and resource intensity of use and health indices are calculated. Intensity of use and health indices integrate quantitative stress from resource use and qualitative diagnoses based on pollution and health assessment. They are used to estimate a composite index of 'internal unit value' for each core account. When no available data, a default value was taken.

Abbreviations: CLC, CORINE land cover; dominant land cover types (DLCT); ESDAC, European soil data centre; GDP, gross domestic product; GPP, gross primary production; IGN, Institut national de l'information géographique et forestière; NEP, net ecosystem production; NLEP, net landscape ecosystem potential; NPP, net primary production; NREP, net river ecosystem potential; RMC, Rhône Méditerranée Corse; SDG, sustainable development goals; SEEA-water, system of environmental-economic accounts for water; SRKM, standard river-kilometre; swbEWA, soil water balance model; TAE, total actual evapotranspiration; TEIP, total ecosystem infrastructure potential.

and statistics are compiled from biophysical characteristics for each of the thematic accounts. In addition to SEEA's land-cover units (i.e., 'ecosystem accounting units' or 'assets'), ENCA defines 'socio-ecological units' which are the complexity level at which ecosystem capital accounts integration can be carried out and ecosystem degradation assessed. As for the SEEA, reporting units can be administrative or geographical divisions.

Land cover and river extent provide the frame defining the statistical units of the accounts (Figure 1b). The CLC inventory of land cover in 44 classes (1-ha pixel-size) for 2000, 2006 and 2012, and change were used to generate dominant land cover types (DLCTs). The inventory also helped to delimit the Rhône basin and hydrological sub-basins.

The combination of DLCTs and river basin boundaries defines the SELUs, the basic statistical units of ENCA (Table 2 and Figure 1c). SELUs allow for the integration of landscapes and the rivers that connect them though their run-off. The SELU has been designed to assimilate and compile input data of different sources and types (spatial resolution, geographic data, statistical data, etc.). Thus, the production of the accounts is done at the object level. We used land-cover map datasets from CORINE (2017) to analyse land cover and change.

Once the land cover and river extent frame has been defined, quantitative and qualitative data are organised for each core account.

Quantitative tables record (Table 2)

1. Quantities of the resource stocks and flows (basic balance),
2. Accessible resource surplus (computed from stocks and flows), which is the resource accessible without depletion,
3. Total uses by economic sectors and

4. Indices of the intensity of use.

Qualitative elements are used to diagnose the health state for each thematic account summarised in an index (see below and Table 2).

In general terms, the production of an integrated assessment of the 'capability' (or potential) of ecosystems in a standard unit, the ecological capability unit (ECU), is calculated per SELU with the available data as follows:

$$\text{Total ecosystem capability} = \text{UVW} + \text{UVC} + \text{UVEI}.$$

The total ecosystem capability is the sum of the ecological unitary value of water (UVW), bio-carbon (UVC) and ecological infrastructure (UVEI), expressed in ECU (Figure 1a):

These unitary values are defined by:

$$\text{Unitary value} = \frac{\text{Health index} + \text{Intensity of use}}{2}.$$

The health index is an assessment of the integrity of the ecosystem, based on intermediate indices for water quality, biodiversity change, and other vulnerability factors with values ranging from 0 to 1

and

$$\text{Intensity of use} = \frac{\text{Accessible resources}}{\text{Use}},$$

where values larger than 1 are set equal to 1 which means no resource depletion due to use. Values between 0 and 1 correspond to situations where use exceeds renewal.

Finally,

$$\text{Accessible resources} = \text{input} - \text{output} - \text{correction factor}.$$

The correction factor is needed because not all the resources are exploitable (e.g., restrictions of use in natural protected areas).

### 2.4 General sequence of account production

The preparatory work consisted in:

1. Spatio-temporal data collection (e.g., global, national, regional, municipal).
2. Data assimilation and integration preprocess (e.g., different tools according to the type of data, OpenOffice, Excel, QGIS, SAGA-GIS).
3. Spreadsheet programmes (e.g., OpenOffice or Excel).
4. Analysis (e.g., Postgis/PostgreSQL spatial data server and calculator Excel).

For example, the ENCA protocol includes data derived from satellite images of Earth and other maps, meteorological and hydrological data, soil maps, biodiversity monitoring data, agriculture and forestry statistics, population censuses and administrative registries. Software and geomatic treatment details are shown in Supplementary Material S3.

We then applied the following sequence.

Collecting data relating to the geographical information infrastructures necessary for the implementation of the method: delimitation of (sub)watersheds, reliefs, rivers, roads, administrative boundaries.

Collecting and organizing various sets of data (i.e., the detailed database) for accounting, namely

1. Socio-economic and environmental statistics,
2. Land cover, as land use defined by the CLC nomenclature and indexed by DLCTs (Figure 2a),
3. Bio-carbon (measured for total vegetation cover and soil carbon),
4. Water resources and
5. Ecological infrastructures.

Creation of geospatial data for each item of the thematic accounts. Data are converted to grids (rasters) of the same pixel size (1-ha pixel-size) to facilitate the calculations needed for the accounting. The processing makes it possible to aggregate the results at the level of predefined natural or artificial entities (e.g., SELUs, sub-basins, municipalities) (Figure 1b and Supplementary Material S4a).

Calculating and editing the accounting tables resulting from the extraction of information from the spatial database. Establishing an ecological balance sheet(s) in physical units.

## 3. Results

According to the accounting framework illustrated in Figure 1, this work reports on:

1. Achieving an in-depth description of the watershed biophysical resources (e.g., accessible resources description, production, supply and the intensity of their use). The framework connects to socio-economic statistics through crop harvests, timber removals, and so forth.
2. Establishing an ecological balance sheet in physical units and aggregating accounts in sequenced steps,
3. Computing synthetic indices of intensity of use and ecosystem health and integrating landscape and river systems at the

SELU scale makes it possible to combine their assessment in the account of ecosystem infrastructure integrity and
4. Establishing the magnitude of change in the 12-year period (tables, graphs and maps) to define patterns and trends at the SELU scale. For each account, areas showing the highest rates of ecosystem capital degradation (i.e., where resources extraction surpasses renewal) are designing potential critical areas or hotspots.

### 3.1 Land-cover stocks and flows

Land-cover stocks correspond to the physical areas of different land-cover types. The land-cover flows account for land-cover changes, that is, conversions between land-cover categories over time grouped according to land-use drivers. They include initial land-cover losses for each land-cover type (consumption), new covers (formation) and the balance between consumption and formation. According to the initial and final land-cover type, the flows are organised in a transition matrix, making possible to generate all possible combinations. Their classification synthetically explains the kind of change between land-cover types (e.g., artificial development over agriculture). Figure 2 summarises the land accounts, focusing on subtle trends in land-cover change by SELU.

A diversity of land uses characterises the watershed, representing three dominant land-cover types: forests (more than 40%), pastures (almost 30%) and arable land (15%). The observed changes in land-use categories for the period 2000–2012 were relatively small and did not change the percentage of the main land-use categories (Figure 2a). However, cartography (Figure 2b) and data analysis (Figure 2c,d) show a sprawling effect due to the following drivers:

1. Land-use change, most prominent in agriculture dominated areas along the Rhône and Saône rivers, and in the northern and southern parts of the territory.
2. A continuous increase in artificialisation.
3. Forest/shrub translation flow effects due to combined losses in agricultural, pasture and forest land (e.g., changes resulting in loss of bio-carbon).

Over the 12-year period, artificialisation increased by 11%, from 4,666 to 5,163 km$^2$, essentially to the detriment of agriculture and pastures (86%), forest (9%), other natural land cover (4%), for example, loss of fertile soils in peri-urban areas. In absolute values, agricultural and pasture land suffered a net loss of 275 km$^2$. The degradation of forested areas (233 km$^2$) led to an increase of transitional woodland (196 km$^2$). Forest degradation has been predominant over restoration activities by approximately a factor of 100. Road artificialisation had impacts on ecosystem infrastructure. A decrease of Glacier area by 13% over the short period considered indicates the rate of climate change effects.

In summary, the reported land-use changes reflect the impacts of conventional agriculture practices in combination with the artificialisation of peri-urban areas and the unbalanced effects of deforestation–reforestation processes.

### 3.2 Ecosystem water and river accounts

The purpose of water and river accounts is to synthesise water resource measurement and its use. It explains water and river

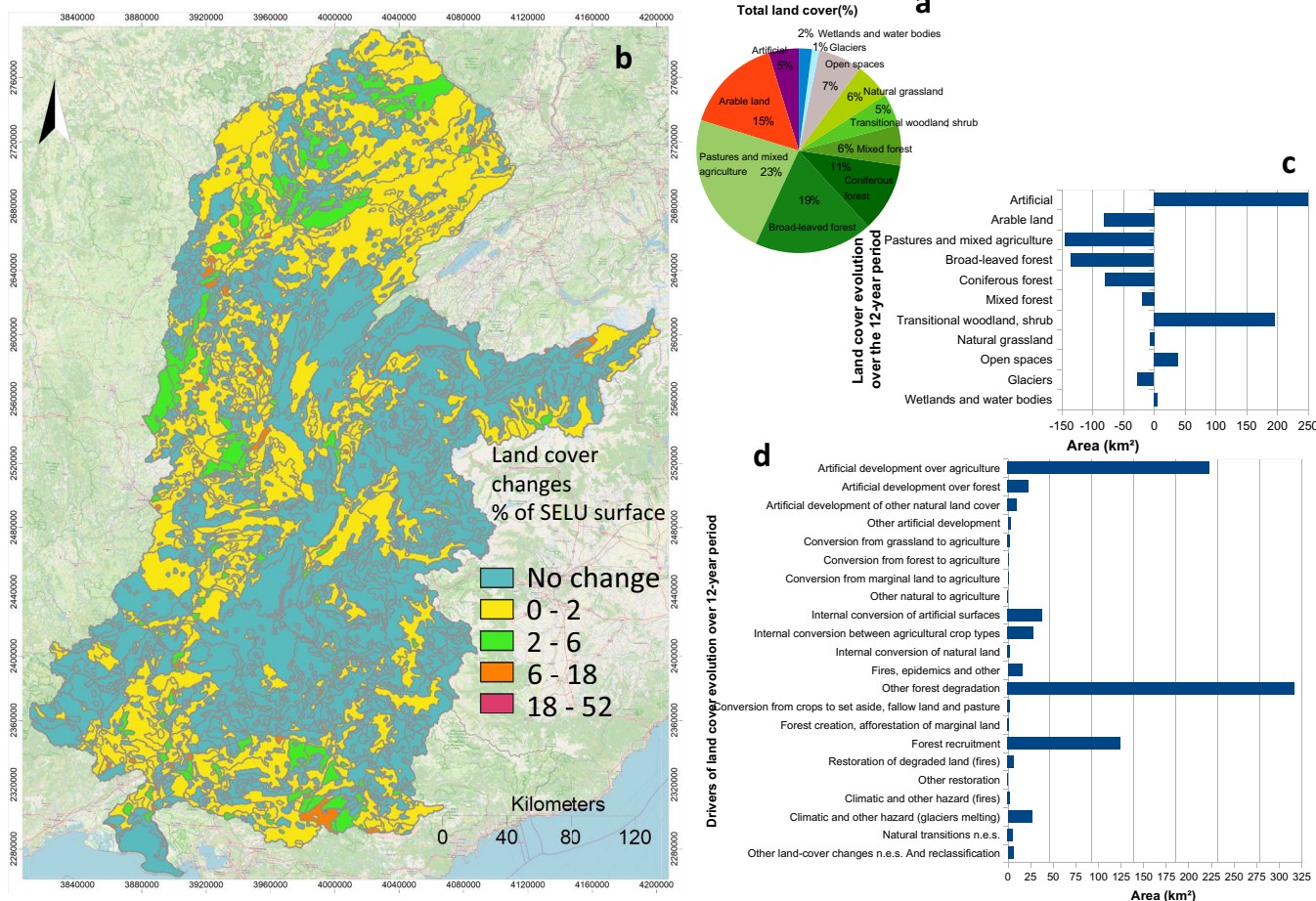

**Fig. 2.** Land-cover accounts. (a) Land-cover main classes coverage (%) of the watershed (CORINE 2000, 2006 and 2012). The CORINE 44 land-cover types were reclassified in 10 land-cover type classes (with an additional distinction between forest categories). (b) Spatial pattern of land-cover surface change in percentage of the surface of the SELU. The shifts between land-cover categories from 2000 to 2012 (CORINE 2000–2006 and 2006–2012; see Figure 1c) represent the summary of the two time periods. (c) Absolute net changes from 2000 to 2012 resulting from the subtraction of formation (gain) minus consumption (loss) of the individual classes per period (CORINE 2000–2006 and 2006–2012) and the addition of the two periods. (d) Drivers of land-cover changes from 2000 to 2012. The drivers of change were classified according to the type of change throughout the period. According to the initial and final land-cover type, the land-cover flows are organised in a transition matrix allowing to generate all possible combinations. The analysis was performed in CLC as a postGIS object using PostgreSQL queries. The abbreviation 'n.e.s.' stands for not elsewhere specified.

networks in the broader socio-ecological sense. An example is the interacting hydrological and user systems within the reference territory.

The definition and classification of statistical units (detailed in Table 2) comprise:

1. River classification (large, medium and small rivers based on their flows).
2. Catchment units at two scales, namely ENCA Catchment or ENCAT (the sub-basin units used for integrating land and river accounts) and elementary Hydrological Zones or UZHYD (the subdivisions of ENCATs), combined with administrative zoning.
3. The intersection of ENCAT boundaries of river basins with DLCTs to define SELUs for river ecosystems (Supplementary Material S4a).

A summary of the structure of ENCA water accounts is shown in Supplementary Material S4b. The main accounting categories are presented below; the definitions and calculation of main indicators being shown in Table 2.

*Quantitative water accounts* record exchanges between the hydrological system units, coupled with the use system of water withdrawals, consumption and returns. The results are compiled using administrative data and statistics withdrawal or estimated through population statistics multiplied by inhabitant equivalents.

*River accounts* closely connect to water accounts by measuring river reaches in *standard river kilometres* (SRKM), defined as the product of their length by the discharge (System of Environmental-Economic Accounts for Water, 2009). With SRKM, weighted rivers can therefore be aggregated.

The results on water stocks, water accessibility, water use and index of water intensity of use are shown in Figure 3. The complete water-river resource was compiled as assets by hydrological units (ENCAT and UZHYD) and the accounts reported by SELUs. The results indicate that:

1. The watershed is a dense, uniformly distributed hydrological system with the Alps, the Massif Central and the Cevennes serving as water 'towers'. The precipitation regimes have been relatively stable over the studied period.

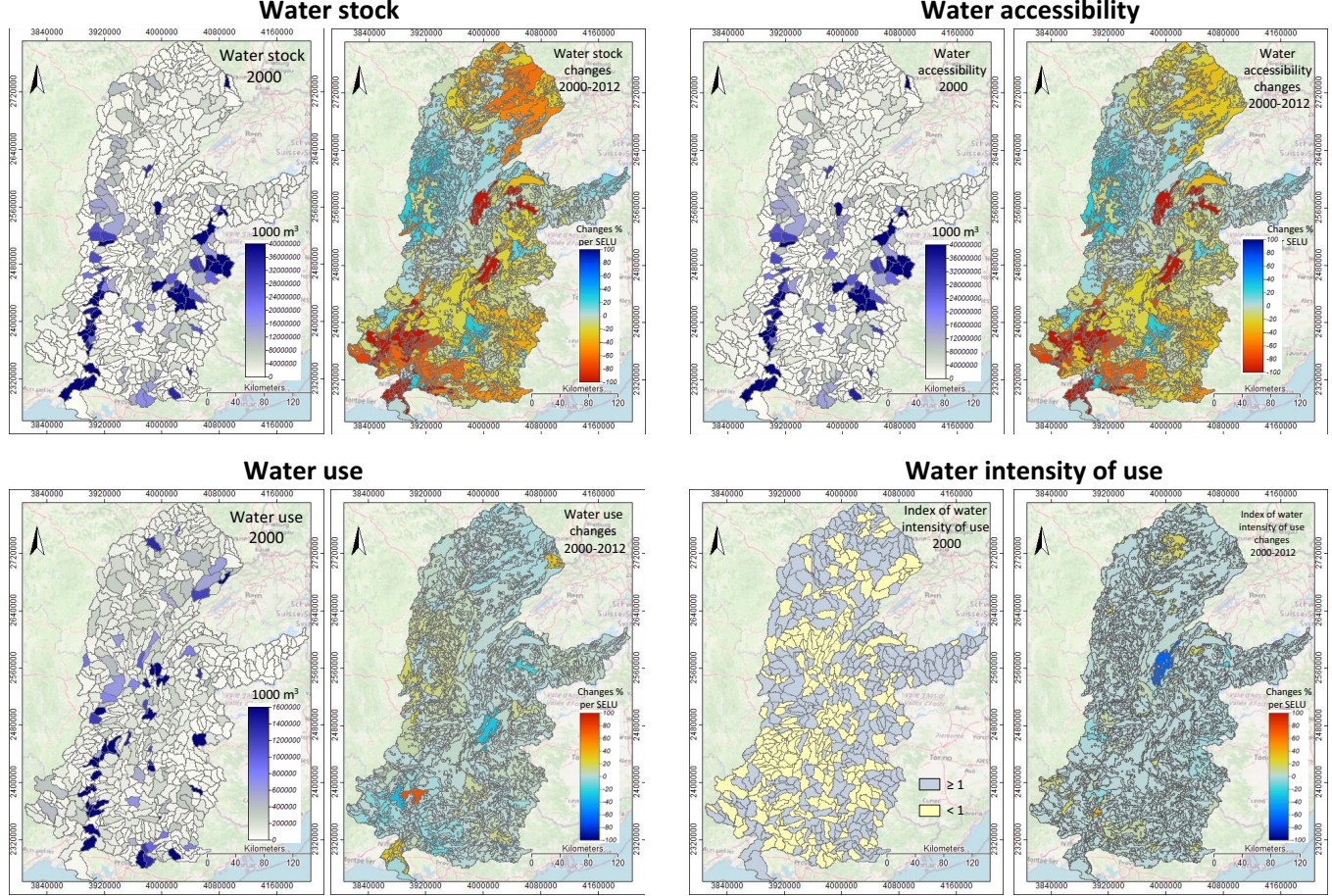

**Fig. 3.** Water accounts by hydrological units (UZHYD) and changes by SELU. Water balanced stocks include estimating water volume in lakes, rivers, glaciers, underwater, soil and vegetation and estimation of outflows (e.g., evaporation, run-off and transfers) and inputs (e.g., precipitation, inflows). Water accessibility reflects the balance of stock and flows, including access restrictions. Water use consists of municipal water withdrawal, agricultural and power production. The index of water intensity of use represents the surplus of water divided by use. In addition, an index has been introduced to account for groundwater stocks that are not measured per se. This approach corresponds to hydrogeology practices, such as measuring the 'piezometric level' and reporting, according to (Water Framework Directive) 2020. The colour code in the right panels indicates the 2000–2012 direction of change: warm colours indicate increased pressure.

2. With relatively predictable water stocks, the territory has agriculture as the primary water user and shows heterogeneous and patchy patterns of change in water accessibility and intensity of use. While the average change in the watershed for water use and water intensity of use is respectively 1.5% and 0.6%, changes in the patches can be up to 40%. This is more pronounced on the Rhône river below Valence, in the drier southern part of the watershed, including the Rhône delta. Hotspots have been identified in the Dombes area (integrated agriculture and pisciculture, and industrial husbandry), south of the Léman Lake, the Chambery–Grenoble couloir, and the southern part of the watershed.

*Water quality* and *River ecological status* (or potential) assessments are based on biological, physicochemical and hydro-morphological quality elements (see also Section 4). They are compiled from information reported by member States to the EU Water Framework Directive (Water Framework Directive, 2000) and the EEA technical note (European Environmental Agency, 2012).

The river ecological status decreased over the studied period (Supplementary Materials S5b and S6e,f). For example, small rivers showed degradation with rates ranging from 5 to 15%. For the main drains, the rates varied from 14 to 20%. While main drains accumulate pollutions from various origins, the network of small rivers is impacted by local pollutions (from point sources or in most cases from agriculture). Visual comparison of impacted areas and land cover clearly shows for example that vineyards north of Lyon are concerned. It contrasts with small rivers in mountain areas where rivers ecological state has improved during the same period.

In summary, the agricultural system constitutes the main source of heterogeneity in water use and the main factor of river ecological status degradation for small rivers in particular.

### 3.3 Ecosystem carbon accounts

So far, research on carbon accounting has explored the subject as discrete inputs and outputs (Nature Portfolio, 2018). No aggregation of carbon accounts on stocks, flows and use in biophysical values has been reported. In ENCA, bio-carbon is measured through the ecosystem's capacity to produce biomass (crops, animal and timber biomass; converted into tons of carbon), combined with use withdrawals and losses. The accounting items, indicators and calculation formulas are given in Table 2 and Supplementary Material S2.

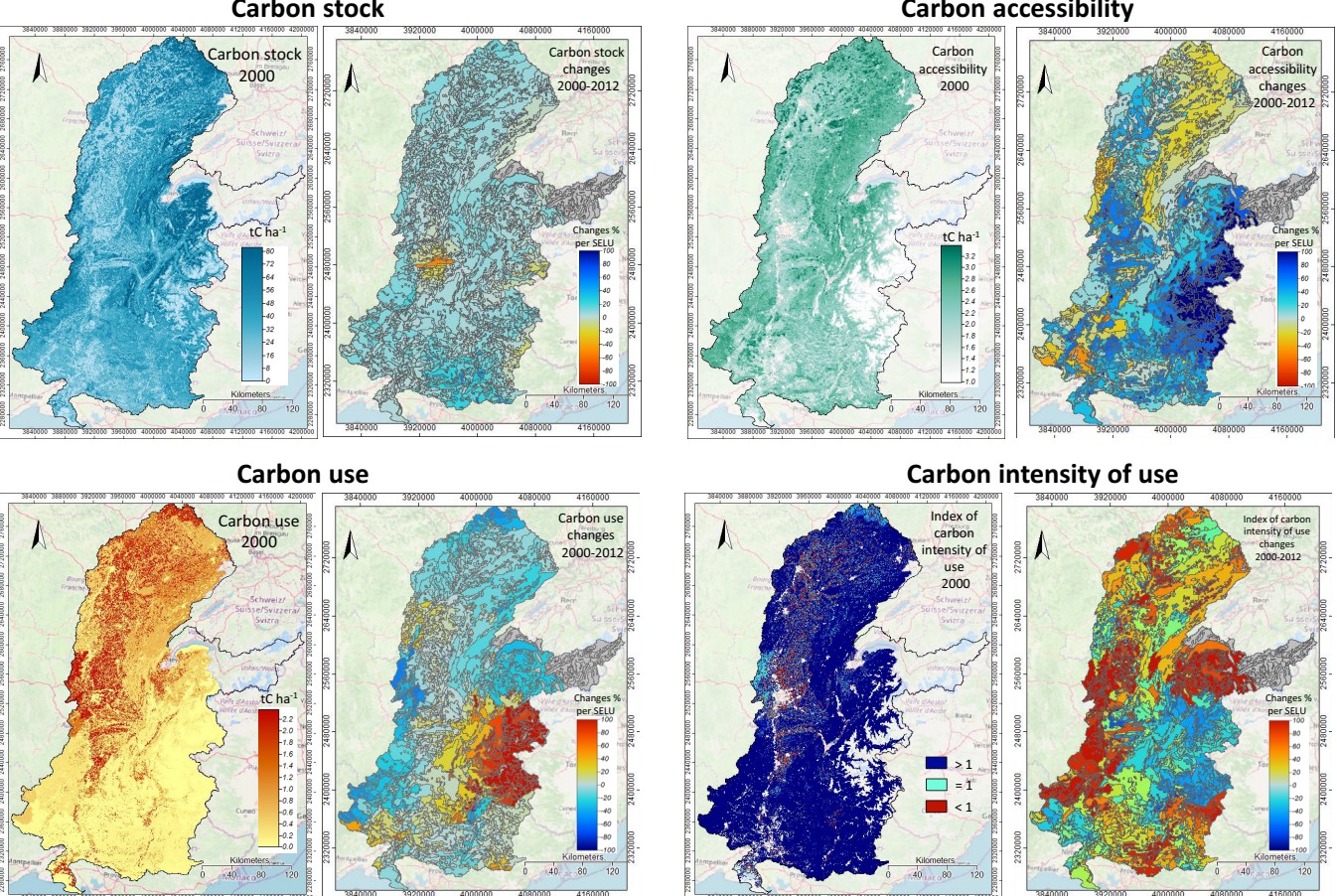

**Fig. 4.** Bio-carbon accounts. Left panels—Carbon stocks (tC ha$^{-1}$) by integration of data on forest (Forested surface and Timber volume from IGN data, and Tree canopy cover percentage from Hansen et al., 2013), deadwood (Forested surface and Timber volume from IGN, and Tree canopy cover percentage from Hansen et al., 2013) and soil (INRA, 2017). Carbon accessibility is calculated as the difference between net primary production (NPP) and heterotrophic respiration (data from NASA-MODIS, 2000–2014). The resulting net ecosystem production (NEP; tC ha$^{-1}$) defines the accessible resource. Carbon use (tC ha$^{-1}$) is calculated as the addition of agriculture harvested crops (from IGN), wood removals (from IGN) and withdrawals of animals (cattle statistics from AGRESTE and pasture class from CORINE). The index of carbon intensity of use in 2000 is calculated as NEP divided by use; values less than one indicate that the use exceeds ecosystem production, that is, degradation. The right panels show the corresponding changes (%) from 2000 to 2012. QGIS and SAGA-GIS geo-processing tools were used. The colour code in the panels indicates the 2000–2012 direction of change: warm colours indicate increased pressure.

This ecosystem approach centres on Net Ecosystem Productivity and Net Ecosystem Carbon Balance (Schulze, 2006). Thus, ecosystem bio-carbon accounting categories cover a large scope of the provisioning ecosystem services and most variables used for estimating human appropriation of biomass (Haberl et al., 2007).

The results are illustrated in Figure 4 and show that bio-carbon stocks are relatively stable over time (with an average of about 37 tC ha$^{-1}$), with forests (trees) as the main resource (90%). There have been considerable variations in bio-carbon use and the intensity of use, showing increase of respectively 10 and 4.5% on average in the watershed, with sprawling patterns along the Saône and Rhône rivers over the studied period. The average percentage of bio-carbon use with respect to the accessible resource has been estimated at 30–40%. There has been an increase in the intensity of use over a third of the watershed (orange areas), most likely due to differences in precipitation-dependent Net Primary Production estimations.

We have identified the following hotspots:

1. Stocks, between Vienne and Valence agriculture couloir, where the loss of the stock ranges from 20 to 50%.
2. Accessibility, north and northeast areas through the Jura, Doubs and Haute–Saône within the limits of the Rhône,

Saône and Loire rivers; south areas through the Drôme, Ardèche and Gard basins.
3. Use, through the Alps-Vercors areas mainly, with the percentage of bio-carbon use with respect to the accessible resource ranging from 32 to 42%.
4. Intensity of use, between Mâcon and Avignon, south of Léman lake, and Gap/Durance areas, with an index increase of 50%. The index indicates that agricultural production in some areas in the Saône river basin is not sustainable (<1, warm colours).

In summary, on a territory with large bio-carbon stocks generated by forests, the major pressure on above and below ground bio-carbon derives from agricultural appropriation of biomass. Soil bio-carbon in the watershed deserves particular focus because it remains an insufficiently characterised resource.

### 3.4 Ecosystem infrastructure accounts

More precisely, the 'Ecosystem Infrastructure Functional Services Account' relates to intangible services that can only be quantified indirectly, given people's access to the ecosystem. It is based

## Net Landscape Ecosystem Potential (NLEP)                    ## Net River Ecosystem Potential NREP

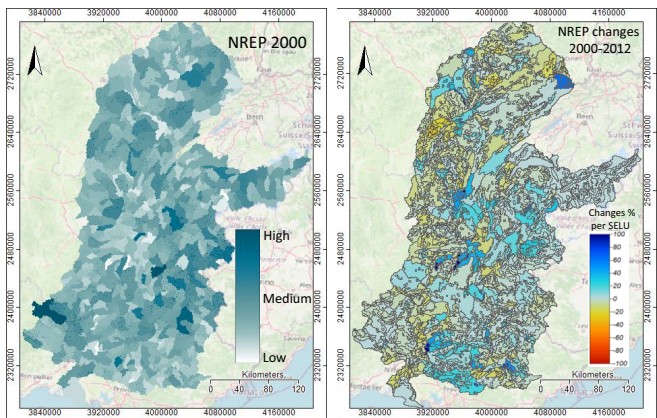

## Total Ecosystem Infrastructure Potential (TEIP)

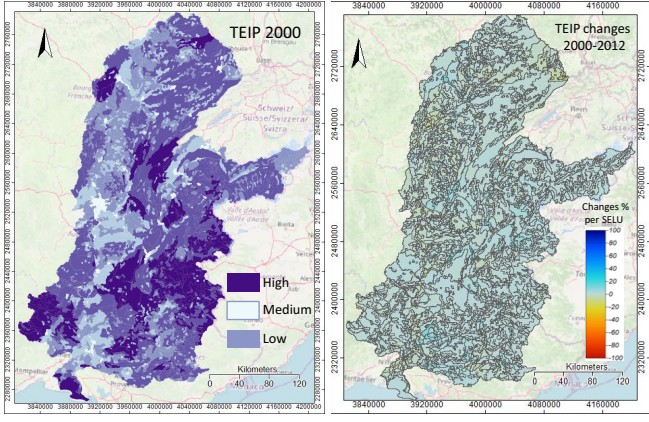

**Fig. 5.** Ecosystem infrastructure accounts (see also Supplementary Material S5). The net landscape ecosystem potential (NLEP) combines an index of greenness, an index of landscape fragmentation and an index to capture the high nature value of particular ecosystems. The net river ecosystem potential (NREP) combines the river condition potential (see Supplementary Material S5b and Supplementary Material S6f) and the index of natural conservation value for rivers. The total ecosystem infrastructure potential (TEIP) is the aggregation (sum) of NLEP and NREP by SELU. The intensity of use for NLEP, NREP and TEIP is calculated as yearly change respectively. The colour code in the right panels indicates the 2000–2012 direction of change: warm colours indicate a decreased potential, that is, an increase in pressure.

on quantitative, semi-quantitative and qualitative estimations of variables. They evaluate the potential access to ecosystem services and impacts on ecosystem functions through the integrity of the ecosystem infrastructure and the ecosystem health within land and river landscapes.

The operating frame is in Supplementary Material S5a–c and indicators definitions and calculations are shown in Table 2. The account aims at producing an assessment of net landscape ecosystem potential (NLEP) and net river ecosystem potential (NREP), based on synthetic indicators designed to characterise the integrity status of various land-cover types and river ecosystems at landscape resolution. The NLEP and NREP indices describe the evolution of such potential. SELUs aggregate the NLEP and NREP indices to compute the total ecosystem infrastructure potential (TEIP) and any changes in the respective intensity of use values.

The results are illustrated in Figure 5 and show that:

1. The NLEP is relatively stable over the analysed period, with a loss of potential of 2% along the Saône and Doubs rivers in the northern part, and of 3% along the Rhône and Durance rivers in the southern part of the watershed (agriculture areas mainly). We identified a hotspot on the Doubs (Basel–Montbéliard area) due to land-cover changes and the resulting fragmentation.

2. The NREP has a similar pattern to NLEP. Still, the changes cover larger SELUs areas than for NLEP and show either improvement (mainly in forested and pasture areas in the southern part) or degradation (mainly in agriculture dominated areas of the northern part of the watershed). It suggests that even small rivers in agricultural areas have a degraded condition (see Section 3.5), but we did not identify hotspot areas.

3. The TEIP reflects the combined degradation effects of land and river condition in agriculture-dominated areas (northern part more broadly, and along the Rhône valley in the southern part). The range of TEIP degradation was estimated at 0.5–30%.

In parallel, the ecosystem health analysis has been performed through the biodiversity index, and the results are illustrated in Supplementary Material S6a. We used this index to evaluate ecosystem health compared with other approaches reporting on biodiversity status on the territory (Supplementary Material S6b–d). Our results on landscape connectivity (Supplementary Material S6d, as defined in Supplementary Material S5b) allow to best capture and visualise relatively discrete fragmentation processes, suggesting that the index is a suitable proxy for biodiversity

**Table 3.** Preliminary integration of thematic accounts of the Rhône river watershed based on intermediate indices of resource intensity of use and health (see Section 2.4)

| Ecosystem account | Changes in intensity of use (%) | Health index (%) | Unit value |
|---|---|---|---|
| Water and rivers | 0.5 | 0.01 | 0.95 |
| Bio-carbon | 4.5 | nd | 0.95 |
| Ecosystem infrastructure | 0.3 | −0.04 | nd |
| Mean | 1.8 | (−0.015) | (nd) |

*Note.* When in brackets, figures are indicative or have not been calculated to avoid inconsistencies. Numbers indicate the average rate of change over 12 years. For the ecosystem infrastructure account, the intensity of use corresponds to the yearly change of TEIP, including fragmentation. The ecological health assembles chemical, biological and functional parameters. Ecological unit values are computed for each thematic account. They are derived by averaging intensity of use and ecosystem health indices (see Supplementary Material S5c for additional components of an ecosystem infrastructure).

Abbreviations: Nd, not determined; TEIP, total ecosystem infrastructure potential.

assessment. Our maps highlight a widespread and continuous degradation in species biodiversity in the French Rhône river catchment, with few exceptions in the Alps. Health index values below 1 signify degradation.

The information on land systems remains insufficient to correctly record damages of pesticides and other chemicals on ecosystem health. However, the Water Framework Directive assessment of rivers' ecological condition (Section 2; Supplementary Material S6e and f) provided valuable information.

In summary, these observations indicate that:

1. The used landscape parameters provide a first hint to evaluate trends in biodiversity.
2. The relative coherence of ecosystem integrity and biodiversity evaluations, and the analysis at SELU scale capture the gradual degradation of increasingly fragmented ecosystems and the reduction of their potential to provide services.

### 3.5 First integration level of the ENCA accounts

The ENCA accounting framework has been articulated in land cover and river ecosystems. This was expected to reveal so far unexplored interdependencies between resource categories in the studied territory. The Rhône river watershed is a geographically diverse territory with important stocks of land, water, bio-carbon and a large diversity of ecosystems. Land-use patterns of change are impacting the bio-carbon balance through high levels of human biomass appropriation and contribute to the fragmentation of landscapes. Increasing rates of water intensity of use and conventional agriculture practices are the main drivers of the ecological state change affecting the rivers' condition potential. Examples are shown in Supplementary Material S6e–g.

Taken together, the results contribute essential spatial information of ecosystem resources with quantified sprawling or patchy patterns of change at the landscape scale. This underlies the diffuse and continuous erosion of all categories of ecosystem capital and delivers early warnings of resource degradation or overuse mainly in agriculture-dominated landscapes.

These results objectify the functional interdependences among accounts. The combined impacts on carbon, water and ecosystem infrastructure resources can now be used to assess changes in the ecosystem potential. The ENCA protocol provides thus a first level of integration through intensity of use and health indices that are thematic account specific but expressed in standard (common) metrics (Figure 1a,b; Section 2.4). The aggregated indices are shown in Table 3.

The results reflect a constant increase in the intensity of use of ecological assets, with a particular degradation of the bio-carbon balance. The average ecosystem infrastructure appeared relatively stable over the 12-year period, for two reasons. First, the network of protected areas combined with limited access conditions has contributed to maintaining the ecological potential. Second, degradation was observed when the analysis was performed by DLCTs, for wetlands in particular, along with water bodies, forests and pastures (data not shown). In general, the picture underestimates the watershed's actual resource base condition due to limitation in available data and information. The gaps for bio-carbon health index and the ecological unit value for ecological infrastructure demonstrate this.

## 4. Discussion

The proof-of-concept work allowed to test and illustrate the different stages of the production chain and to identify the data required for its implementation. Based on the reported results, the question is: How does ENCA compare within the current composite and scattered methodological landscape of ecological capital evaluations?

To our best knowledge, no direct comparisons between existing methodologies have been performed so far. To achieve an in-depth understanding of their respective strengths and limitations, such comparisons would need using the same sets of data on the state and dynamics of the ecosystem capital across countries or contexts. It is likely that the required data may differently fit the specifics of such methodologies and, once performed, in context validation of the results would take more time than expected (as in this work on distinct data systems between France and Switzerland). Additional limitations need to be mentioned:

1. Methodological concerns, as for the ecological footprint (Blomqvist et al., 2013).
2. A restricted choice of assets with as result partial accounts (Australian Bureau of Statistics, 2013; International Institute of Sustainable Development, 2018; Ouyang et al., 2020; Wigley et al., 2020), and corporate reporting on biodiversity (based on Life Cycle Assessment and Pressure-State-Response proxies; Delavaud et al., 2021).
3. Indicator systems still in development (Fairbrass et al., 2020), and in particular those monitoring biodiversity (with some consensus on focusing assessments on ecosystem area, integrity and risk of collapse; Rowland et al., 2019).

4. Aggregation of ecosystem service accounts in money, with preference for measuring flow values rather than changes in stocks (Economics of Land Degradation, 2015; The Economics of Ecosystems and Biodiversity, 2008).

5. Monetary valuation resting on micro-economic principles hardly transposable to national accounts (mainly InVest and Co\$ting Nature tools to model ecosystem services provisioning for case studies; Delavaud et al., 2021; The Economics of Ecosystems and Biodiversity, 2008; United Nations Environment Programme, 2014).

On these grounds, we argue that the best option would be the systematic assessment of the state and dynamics of the ecosystem capital covering its core components, namely land use, and the state of water/rivers, biomass and ecosystem infrastructure. This is what ENCA does, with the aim of engaging various stakeholders and the society at large in the evaluation and the stewardship of their territory. The systematic integration of the required variables in the ENCA framework, their spatial breakdown, cross-checking and update brings more than the possibility of calculating particular indicators such as total ecosystem capability and its degradation. The ENCA database is at the same time a possible source of pre-processed data for a range of other applications. As such, it would provide linkages to the variety of applications or projects. In the same vein, Fairbrass et al. (2020) (see also Ekins et al., 2019) propose a guide for natural capital assessment. The natural capital indicator framework organises a large number of indicators into a coherent structure of key and headline indicators based on the four-capital model of wealth creation (e.g., natural capital stocks of ecosystem and commodity assets, ecosystem flows from natural capital, human inputs and outputs in the form of benefits and residuals).

### 4.1 ENCA proof-of-concept: A first determinant step towards an exhaustive and integrated evaluation of ecosystem capital value

The system of ecosystem accounts approach simultaneously addresses externalities for key primary resources associated with infrastructure and ecological health, including land, water and biomass (see also Negrutiu et al., 2020). By monitoring stocks and flows:

1. Trend values of capital stocks allow forecasting of the future potential of the stocks.

2. The relative value of various flows of ecosystem resources according to their use can be better understood and thus determine the extent of changes needed to address the opportunities for sustainable use.

3. The costs relating to the use and degradation of the ecosystems that are presently unpaid (e.g., restoration, avoidance or compensation costs) can be evaluated.

The Rhône watershed core accounts support reporting on societal targets, test territorial scenarios and catalysing action. The standard metrics and matrix—based on intensity of use and health indices for bio-carbon, water and ecosystem infrastructure—comprise an essential step forward. They enable a more systemic understanding of the territory's strategic resources. ENCA leaves open the possibility of monetary valuation for methodological comparisons and prospective modelling.

Mapping the actual trends of resource stocks, flows and use allows the location of hotspot areas where the drivers of degradation can be identified. Such sites need verification with other sources and local actors to inform trade-offs and activate mitigation through decision-making.

Two examples illustrate the method's strength: land-use and soils, and biodiversity and landscape.

Land cover and land-use changes constitute a fundamental issue, as seen in the overall scale of the process; it is the largest geoengineering human activity of all times (Verburg et al., 2015). Our analysis of land-use patterns and the state of terrestrial ecosystems in the watershed shows that it is the primary driver of ecosystem fragmentation. With a reduction in ecosystem diversity and productivity come habitat loss and degradation, biodiversity erosion and bio-carbon balance disruption (see also Barnosky et al., 2012; Steffen et al., 2015; Urban, 2015; Verburg et al., 2015).

The study targets questions such as:

1. How to maintain the quality and integrity of the land stock through land management (Haines-Young et al., 2006) to provide a better integration of soil bio-carbon and land use. There is an urgent need to arrest global agricultural land degradation, currently estimated at 67% (Prăvălie et al., 2021).

2. Finding solutions for the unsolved issue of a major disconnect between the financial value of land and land value according to the multifunctional capabilities of land (see also Terama et al., 2019).

This is a driver of artificialisation and while ENCA has not been calibrated yet to assess all impacts of urban areas on ecological capital, urban metabolism studies (Raworth, 2012) can be integrated into ENCA environmental assessments.

Efforts to fix problematic issues are noticeable (Economics of Land Degradation, 2015; International Conference on Computer Design, 2017 on Land Degradation Neutrality; National Academies of Sciences, Engineering, and Medicine, 2021; Verburg et al., 2015). Interestingly, the EU has long-established water and air directives, but no soil directive (claimed through citizen action; European Citizens' Initiative, 2017).

Biodiversity evaluation raises additional but no less critical concerns. There have been continuous efforts to implement a system of conventional biodiversity metrics (Biodiv2050 Outlook, 2017; Diaz et al., 2020; Pereira et al., 2013). Nonetheless, providing near real-time information for systematic and regular biodiversity assessment would be resource-intensive and have questionable relevance for decision-making (Mazor et al., 2018; see also Kwok et al., 2019; Wyborn et al., 2020). To go further than the reporting to the European Habitats Directive (art. 17) (2012), a versatile biodiversity data monitoring to measure impacts systematically should take stock of the fact that changes through habitat structure remote sensing (loss, degradation), fragmentation levels and land-use change identify the primary cause of substantial changes in species abundance, distribution and interaction (Brooks et al., 2002; Dirzo et al., 2014). They can be monitored simultaneously and globally (Mace et al., 2014). The ecosystem approach (Rowland et al., 2019) has been designed to develop indicators on ecosystem area and integrity assessment.

This is what ENCA does. Our results show that ecosystem infrastructure and ecosystem health assessments at landscape scale proved consistent (Figure 5 and Supplementary Material S6d) compared to alternative approaches (Supplementary Material S6b and c). The ecosystem infrastructure account is an actionable proxy allowing researchers and society actors to target areas of actual or

**Table 4.** Obstacles (coloured cells) in the production of Rhône river watershed ENCA accounts

| Obstacle designation | Land use | Bio-carbon accounts | Water account | Ecological infrastructure and synthetic indicators | Summary scores |
|---|---|---|---|---|---|
| *Analysis of stocks and flows; definition of spatial entities* | | | | | |
| Grouping of land-cover classes | | | | | 1 |
| Grouping of stocks and flow categories | | | | | 1 |
| Designing flow classes versus vegetation types | | | | | 1 |
| Spatial unit designation | | | | | 3 |
| *Data sources management* | | | | | |
| Satellite data limitations | | | | | 3 |
| Data sources accessibility | | | | | 3 |
| Lack of time series | | | | | 4 |
| Incomplete (gaps) and quality of data | | | | | 5 |
| Errors in the source data not explicit | | | | | 3 |
| The disparity of data providers | | | | | 1 |
| Heterogeneity of data | | | | | 3 |
| Scale and interoperability | | | | | 3 |
| *Analytical instruments and tools* | | | | | |
| Heterogeneity of geoprocessing software and tools | | | | | 4 |
| Geoprocessing errors | | | | | 1 |
| Indicators and indices. Scoring criteria and statistical models | | | | | 2 |
| Assumption and normalisation | | | | | 3 |
| *Score (per account category)* | 6 | 11 | 11 | 13 | |

*Note.* Obstacles were classified into three categories and broken down by each type of account: land cover and use, bio-carbon, water and rivers, and ecological infrastructure. Pale yellow indicates the presence of obstacles in the production of the accounts with indirect impact. Yellow indicates the presence of obstacles with direct impact. Red indicates a combination of direct and indirect impacts. Orange shows that the data for the water accounts were fairly abundant and of good quality, but we encountered problems with water use, management and distribution. The final score per column ranks the accounts according to the technical difficulties encountered.
Abbreviation: ENCA, ecosystem natural capital accounting.

potential biodiversity erosion due to land-use change and unsustainable practices. These are areas where focusing on more detailed biodiversity assessments is required (Plumptre et al., 2021). In brief, using the landscape scale in biodiversity assessment can capture the dynamics of the process with a reasonable accuracy simultaneously and world-wide.

### 4.2 Data resources: The main extrinsic obstacle in deploying a complete ENCA proof-of-concept

The colour code in Figure 1a illustrates how the sequential accumulation of data limitations has been a handicap in generating aggregation indices (such as intensity of use and health indices) and working out a complete ENCA proof-of-concept. This has been the main reason for not aggregating the thematic accounts through the ecological value and capability metrics (compare Table 3). Table 4 inventories the obstacles—namely gaps and significant heterogeneity in geographically localised data production, software and processing tools—in data and information source management. It also indicates which steps in the ENCA procedure suffer most from data limitations and defines spatial entities and inconsistencies in the scales of restitution. It describes the quality of the statistics on which the measurement of the physical state of ecosystems is based.

For example, for the bio-carbon accounts, flows data need to be completed by including carbon loss, respiration and leaching, disturbances, additional secondary stocks, inflows from other countries, and production and consumption return to the ecosystem at the required scale. For the use variables, agriculture information lacks at-scale spatial information and time series, while data on fisheries do not exist.

In short, no administrative or political authority can provide the data required to assess on a regular and consistent basis the state of the ecosystem capital in the Rhône basin territory for which they are responsible (Auvergne-Rhône Alpes, 2016). Ideally, generating annual series is an objective to reach and capture the combined result of trends and seasonal or annual fluctuations related to the meteorology (such as the Net Primary Production or evapotranspiration and rainfall).

Beyond ENCA-Rhône and despite ever-expanding satellite, sensors, geospatial data production and structuring efforts (POST, 2017), a general trend in data management is the lack of sufficiently robust, systematic and spatially explicit data collection, treatment and access systems (Moran et al., 2020; Natural England Commissioned Report, 2020; Wilkinson et al., 2016). Consequently, results frequently come from aggregated data or surrogate modelling and data extrapolation to remediate inaccurate data and errors. We expect the data to become findable, accessible, interoperable and reusable.

In summary, the main obstacles in data management and science concentrate on two dimensions:

1. Making progress towards converging concepts, definitions and methodologies in the environmental evaluation field across various player groups.

2. Coherent public data policies (e.g., adequately institutionalised data collection and processing) to support exhaustive, reliable and systematic evaluation of the ecosystem capital, enabling the calculation of unpaid degradation costs.

### 4.3 Conclusion and perspectives

Despite the above obstacles, we showed ENCA to be a methodological breakthrough, an instrument exhaustive enough to assess the ecological assets at the watershed scale and operate to monitor early warning signals of ecological capital degradation. Importantly, ENCA core accounts per se constitute already an indispensable record of data in decision-making. Displayed on an interactive dashboard, the device would indicate whether and how the degradation of resource categories spatially overlaps. The stepwise aggregation of ENCA components can provide intermediate-to-global indicators telling whether GDP growth is correlated or not to the degradation of defined resource categories of the ecosystem capital. Such relevant decision-grade information is needed in integrated resource management, landscape planning and scenario building.

At local level, ENCA can help design charters and protocols to meet project targets while helping integrate initiatives and foster partnerships and empowerment. For example, we have produced an inventory of potential partners in the territory (Parmentier et al., 2021) for networking local partnerships. The study has designed the contour of a platform for territory-actors-resources with the aim to

1. Facilitate arbitration and trade-offs in managing available resources,
2. Calibrate public markets and the conditionality of public contracts,
3. Helping businesses understand the state of the ecosystem capital and associated financial risks in the territories in which they operate, and
4. Integrate ecosystem capital accounting in local wealth assessments.

Such a co-construction effort is critical for local actors in better understanding their territory, its specifics and potential and empowers citizens to act and vote on landscape and territorial stewardship matters.

### Acknowledgements

We wish to thank Charlotte Weil for her critical reading of the manuscript and productive suggestions of improvement. We are grateful to the two reviewers for their patience and precious advice in making the article accessible to all.

**Financial support.** This work was supported by the National Council on Science and Technology, Mexico (CONACyT) (Argüello Velazquez Jazmin Adriana, grant number 225397/410110), and the Ecole Normale Supériéure de Lyon, France: the Laboratory of Plant Reproduction and Development, the Michel Serres Institute, and the Rhône-Alpes Institute of Complex Systems.

**Conflict of interest.** The authors declare that they have no known competing financial interests or personal relationships that could have appeared to influence the work reported in this paper.

**Authorship contributions.** J.-L.W. and I.N. conceived and designed the study. J.A. and J.-L.W. conducted data gathering and performed statistical analyses. J.A. and I.N. wrote the article and J-L.W. contributed corrections and improvements.

**Data availability statement.** The data publicly available that support the findings of this study are available from Base de Données sur la Cartographie Thématique des Agences de l'eau (BD Carthage), Wikipedia, Coordination of information on the environment CORINE, version v.18.5, European Soil Data Centre (ESDAC), The soil water balance model (swbEWA), Base de Donnée des Limites des Systèmes Aquifères (BD LISA), Institut national de recherche en sciences et technologies pour l'environnement et l'agriculture (IRSTEA)/Office national de l'eau et des milieux aquatiques ONEMA, WorldClim, French Water Agency Rhône Méditerranée Corse (RMC), Institut national de l'information géographique et forestière (IGN), Global Forest Change 2000–2017 V1.5, Land Use/Cover Area frame statistical Survey (LUCAS), National Aeronautics and Space Administration (NASA-MODIS), Global Database of Soil Respiration Data (ORNL DAAC, NASA), Service de la statistique et de la prospective du Ministère de l'Agriculture, de l'Agroalimentaire et de la Forêt (AGRESTE), Open Street Map (OSM), Inventaire National du Patrimoine Naturel (INPN), European Environmental Agency: Habitats Directive—Art.17. Restrictions apply to the availability of resources on forestry surface and soil carbon stock, which were used under licence for this study.

**Supplementary Materials.** To view supplementary material for this article, please visit http://doi.org/10.1017/qpb.2022.11.

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
