## [Reviewer Report]

Dear Editor-in-chief,

We submit to the QPB journal our manuscript

Ecosystem Natural Capital Accounting. Proof-of-concept - the landscape approach at watershed scale authored by Jazmin Argüello, Jean-Louis Weber, and Ioan Negrutiu.

We thank you for having invited us to contribute this manuscript to the citizen science section. This attribution makes sense because our work has benefited from a broad range of data sources and collective knowledge and expertise. This is outlined in brief below in two steps

- defining the science and society context in which our approach and work has been designed, and

- giving a brief account of the scope, results, obstacles, and perspectives of the work.

Scholars have tried to clarify the value of a healthy ecosystem given the lure of short-term profits from its incremental degradation. Still, we believe the extant body of research on this urgent topic has omitted some significant needs and concerns for natural capital evaluation. 

We surveyed the field to capture current trends in environmental evaluations and classified them according to three main conceptual categories, namely (1) Pressure based indicators: reference value or limits (such as disjoint environmental indicators, ecological footprints and planetary boundaries); (2) Ecosystem services assessment and valuation; (3) Assessment of ecosystem state and degradation in a framework congruent with conventional national accounts.

There is also a lack of public data policy to support sorely needed ecosystem capital evaluation that is coherent, systematic, and exhaustive. Also, there is a need for institutionally-supported methodologies that have been peer-reviewed and are thus grounded in scientific protocols. 

Thus, a scattered array of research goals and methodologies appears to have stalled a political commitment to effective ecosystem and resource stewardship, and generated confusion in the society at large as to the capacity of science to develop and recommend actionable accounting solutions that measure what counts in the general interest while moving on the road toward strong sustainability. 

In this challenging context, we focused our experimental work on the assessment of ecosystem state and degradation (category number 3 above) and developed an ecosystem capital accounting tool for the Rhône river basin, a territory that possesses important stocks of bio-resources. The results are based on geographic spatio-temporal information and statistical units over a time period of 12 years. The study reveals significantly higher physical details on the dynamics of critical resources stocks and flows than previously assessed; namely land use, water and river condition, bio-carbon content of various sources of biomass, and ecosystem infrastructure. They are organized at landscape scale, a socioecological unit with convenient economic, political, and societal significance. We attempt demonstrating that the landscape unit is a suitable proxy in ecosystem capital assessment that will be able to capture the dynamics of territorial resource state with a reasonable accuracy and costs, simultaneously and world-wide.

We discuss obstacles and limitations we have been confronted with while developing the proof-of-concept and suggest improvements that will allow

- upgrading and aligning the existing instruments with the requirements of strong sustainability and the principle of "no net ecosystem degradation";

- facilitating systematic comparisons of the ecosystem capital state in space and time;

- providing a matrix for decision-making supported by a better understanding of ecosystem capital, biodiversity, and nature-society relationship;

- creating conditions to associate and involve all the actors within given territories in resource stewardship through good practice and fair reporting.

We sincerely believe our findings would appeal to the readership of QPB.

We confirm that this manuscript has not been published elsewhere in its present form and is not under consideration by another journal.

---

## [Reviewer Report]

Dear Editor-in-chief,

We submit to the QPB journal our manuscript

Ecosystem Natural Capital Accounting. Proof-of-concept - the landscape approach at watershed scale authored by Jazmin Argüello, Jean-Louis Weber, and Ioan Negrutiu.

We thank you for having invited us to contribute this manuscript to the citizen science section. This attribution makes sense because our work has benefited from a broad range of data sources and collective knowledge and expertise. This is outlined in brief below in two steps

- defining the science and society context in which our approach and work has been designed, and

- giving a brief account of the scope, results, obstacles, and perspectives of the work.

Scholars have tried to clarify the value of a healthy ecosystem given the lure of short-term profits from its incremental degradation. Still, we believe the extant body of research on this urgent topic has omitted some significant needs and concerns for natural capital evaluation. 

We surveyed the field to capture current trends in environmental evaluations and classified them according to three main conceptual categories, namely (1) Pressure based indicators: reference value or limits (such as disjoint environmental indicators, ecological footprints and planetary boundaries); (2) Ecosystem services assessment and valuation; (3) Assessment of ecosystem state and degradation in a framework congruent with conventional national accounts.

There is also a lack of public data policy to support sorely needed ecosystem capital evaluation that is coherent, systematic, and exhaustive. Also, there is a need for institutionally-supported methodologies that have been peer-reviewed and are thus grounded in scientific protocols. 

Thus, a scattered array of research goals and methodologies appears to have stalled a political commitment to effective ecosystem and resource stewardship, and generated confusion in the society at large as to the capacity of science to develop and recommend actionable accounting solutions that measure what counts in the general interest while moving on the road toward strong sustainability. 

In this challenging context, we focused our experimental work on the assessment of ecosystem state and degradation (category number 3 above) and developed an ecosystem capital accounting tool for the Rhône river basin, a territory that possesses important stocks of bio-resources. The results are based on geographic spatio-temporal information and statistical units over a time period of 12 years. The study reveals significantly higher physical details on the dynamics of critical resources stocks and flows than previously assessed; namely land use, water and river condition, bio-carbon content of various sources of biomass, and ecosystem infrastructure. They are organized at landscape scale, a socioecological unit with convenient economic, political, and societal significance. We attempt demonstrating that the landscape unit is a suitable proxy in ecosystem capital assessment that will be able to capture the dynamics of territorial resource state with a reasonable accuracy and costs, simultaneously and world-wide.

We discuss obstacles and limitations we have been confronted with while developing the proof-of-concept and suggest improvements that will allow

- upgrading and aligning the existing instruments with the requirements of strong sustainability and the principle of "no net ecosystem degradation";

- facilitating systematic comparisons of the ecosystem capital state in space and time;

- providing a matrix for decision-making supported by a better understanding of ecosystem capital, biodiversity, and nature-society relationship;

- creating conditions to associate and involve all the actors within given territories in resource stewardship through good practice and fair reporting.

We sincerely believe our findings would appeal to the readership of QPB.

We confirm that this manuscript has not been published elsewhere in its present form and is not under consideration by another journal.

---

## [Reviewer Report]

*Comments to Author*: Dear Jazmin and colleagues,

I reviewed your manuscript entitled « Ecosystem natural capital accounting. Proof-of-concept-the landscape approach at watershed scale” submitted to “Quantitative Plant biology” in the “Citizen Science” section.

The paper aims to propose a new framework and tool (ENCA) to assess the ecological value of an ecosystem, i.e. with no monetary valuation as the majority of studies do to estimate an “ecosystem value”. You described the structure and discussed the advantage for ecosystem management from a pilot study you conducted on the Rhône river.

In my opinion, this change of paradigm from monetary valuation to ecological valuation is fundamental in a global change context, to then propose efficient ecosystem management actions. The ENCA project seems very promising but, for me, your manuscript did not fully demonstrate the potentiality and the power of the project. I give you more details about my reviews and I hope it can help for the future of this manuscript.

Indeed, the task is very hard as ENCA seems considering a lot of factors and processes influencing ecosystem functioning. I feel that you wanted to give a large overview of ENCA but lots of information was missed to fully understand how it works. I would suggest for instance to focus only on one output like water and to detail more precisely how the model works.

In the introduction, I was disappointed that you did not propose a bigger paragraph on the monetary vs ecological valuation, which is for me the biggest added value of ENCA. You did it a bit in the definition section of “ecological value” and I think it would fit better in the intro Moreover, the flow of the reading is sometimes not easy with a lot of bullet points and the link between paragraphs are not obvious. I felt like reading lists instead of a demonstration leading to the need to use ENCA.

Some sentences miss some references or scientific background. For example, when you talked about “ecological debt” and when I read the definition and the source you gave, I am not sure there is a strong scientific meaning behind it. It is a pity since your goal is to give a standardized tool to assess ecological values and in fine maybe quantify these “ecological debts” as you proposed L125. With this definition, it is impossible. Generally, I think your definitions are too broad and conceptual, at the end of my reading, I was not sure what exactly you quantified:

- In ecological value definition: maintenance and resilience may be viewed from an ecological or anthropic point of view even without considering monetary valuation. What do you mean by resilience for instance?

- ~L975 ecosystem assets and services, I would replace services by function, services will certainly lead to a monetary point of view

- “the ecological value… ecosystems functions” was already said before

- “the potential” is not clearly defined in your paper

- “territory-landscape-land cover” did not appear in the text, why do you define it? If it’s something you forgot which representation do you choose?

- “watershed”, I suppose that the primary definition comes from geographers and/or biologists, we need this approach in your definition.

-

The materials and methods gave some clues about your model but the indices you used are not described or very superficially in your table… When we have some details, it is relatively weird for some: for C, Heterotrophic respiration is different from autotrophic respiration (and why did you write NPP, MODIS-NASA?)

I suggest that you propose another figure with the link between the different input and output of your model, to have a graphical description of how ENCA consider each data set and indices. When you used “include”, “combine data set”, we need to know how it is done, what is your “recipe” to then get the maps you showed.

In the discussion, in the section starting L490, I think one interesting point missing is to compare your result with a monetary valuation to show the interest in considering ecological values. Some paragraphs just confirmed what we already know such as the ones ~L505 and 510 and you should highlight the new information provided by ENCA.

Finally, in paragraph ~L595, you discuss citizen sciences implications. There are almost no references to support it and I strongly suggest adding several to exemplify your statements. Moreover, except as a very indirect by-product of ENCA, I did not see the citizen science relationship with the main message of your paper. Then I do not think that your manuscript fits in this section of the QPB journal.

I am quite critical and sceptical after reading your paper because I think the content was too ambitious for a restricted format. The take-home message is relatively clear, i.e. to propose an ecological valuation tool, but the demonstration needs to be more structured and clarified. They're also a lot of acronyms, which do not facilitate the reading (but I understand that it is hard to suppress some of them…). Moreover, please, add numbers to each line to facilitate the reviewing process.

I would like to end with a positive sentence, once again, it is fundamental to build a common ecological metric to characterize the ecosystem state. I think it is timely work and I will carefully follow this project.

Some minor revisions:

~L35: economism? What do you mean?

~L40: what do you mean by “natural capital consumption records”

“corresponds” not only, disturbing network may also lead to loss of ecosystem’s ability

~L45: what is GDP?

~L50 “living in harmony with nature” need to be concretized

~L55: slowed

~L65: West, 2015) AND aimed?

~L445: “the processes often overlap” can you develop? What are the consequences?

~L500: “land-use changeS”

Figure captions:

You should reorganize your choice of figures, it is weird to refer to fig 2a in your fig 1. Also ~L1115, these are some points of results or discussion

~L1125: could you explain what do the scores means (I guess it is linked to the indices but we do not know if a change of one unit is important or not

~L1155: can you explain more precisely what “accessible” means? I would say “ARE accessible”

---

## [Reviewer Report]

*Comments to Author*: Dear authors,

Thanks for submitting a revised version of your manuscript “Ecosystem natural capital accounting. Proof-of-concept-the landscape approach at watershed scale”. In my opinion, and after reading your answer to reviewer 2, the manuscript has been improved.

My main concern is still about the citizen science section of the journal and your paper content. In my sense, this paper does not fit at all in the citizen science section of the journal. On the journal website, the scope is “Citizen science papers: building on very large datasets across wide regions from non-scientists, co-written with scientists” which is not the case o your paper. It does not mean that your paper should not be published, but maybe not in this section. This is my opinion and I leave the decision for the editor.

I’m still sceptical about “living in harmony with nature”, the SDG 2020 and CBD 2018 used this term but they do not define it (except if I miss it). I really think you should reformulate, as it is, it is relatively a hollow term. Moreover, the CBD 2018 reference is a presentation, I’m not sure the content really helps to support your claim, very little information are present on the slide (the talk would have helped but unfortunately there is no file like that).

L86: SEEA appears for the first time in the paper but it is only full worded L88.

Fig S1: can you explicitly write in the caption that blue lines represent the departments’ borders and that red lines represent the watershed

Fig S2a: you have different items in French, is there any reason? It would be better to translate it into English.

Fig S2b: can you improve the quality of the picture, first lines are for instance, unreadable.

Fig S4: what do the colours mean? Can you put a name above the 3 columns in figures b and c.

I think the sentences mentioning sections 3 and 4 L293 and L300 should be removed from section 1.

---

## [Reviewer Report]

*Comments to Author*: The text has been improved significantly. There are still some items to clarify (mainly the definitions of the indices), and the discussion and figure legends are also too long. Overall, the message is better conveyed now, and with this last set of corrections, the main take home messages should go through.

Suggestions for changes in the abstract:

“Abstract:

Most approaches to estimate ecosystem value use monetary valuation. Here we propose a different framework accounting ecological value in biophysical terms. More specifically, we are implementing the Ecosystem Natural Capital Accounting (ENCA) framework as an operational adaptation and extension of the UN System of Economic and Environmental Accounting/Ecosystem Accounting (SEEA-EA). The study was carried out at the Rhône River watershed scale (France). Four core accounts evaluate land use, water and river condition, bio-carbon content of various stocks of biomass and its uses, and the state of ecosystem infrastructure. Integration of the various indicators allows measuring ecosystems overall capability and its degradation. The 12-year results are based on spatio-temporal geographic information and local statistics. Increasing levels of intensity of use are registered over time, i.e., the extraction of resources surpasses renewal. We find that agriculture and land artificialization are the main drivers of natural capital degradation. »

Line 55 “Two such levers, namely the diffusion of strong sustainability notions (Table 1) and the emergence of environmental evaluation thinking and practice have been slow to gain ground. » I think the authors simply means that “One such lever, namely the development of strong sustainability (Table 1) building on the evaluation of ecosystem services has been slow to gain ground.”

Line 57 “Environmental evaluations, essential tools to socioecological systems approaches (Bourgeron et al., 2018; WWF, 2019; Li et al., 2020), are being developed to assess the direct or indirect impacts of externalities on ecological systems and their productivity. » This sentence should be embedded in the following paragraph, which deals with the different methods.

Line 67: Please consider the following rephrasing:

“A significant reason is that political and economic decision-making and societal choices are restricted by the following limitations or contradictions in the current instruments (Arguüello et al., 2020):

(i) Ecosystems are often reduced to their monetary value and are aggregating distinct categories of ecosystem capital thus hindering other possible frameworks.

(ii) Methodologies often target the “intensity of resource use” and measure flow values rather than changes in stocks.

(iii) Methodologies attempting to encapsulate the GDP within more or less strict ecological limits and sustainability are heterogeneous

(iv) Ecosystem services assessments are confronted with the challenge of the interconnected and multifunctional nature of services (e.g., avoiding double-counting or incomplete services counting or none at all).

Here, we implement a novel approach, called Ecosystem Natural Capital Accounting (ENCA), which instead considers accounts in biophysical terms. »

Line 89: “However, regarding monetary assessments, the SEEA-EA cornerstone is valuation of the benefits provided by ecosystem services and assets, whereas ENCA approach to biophysical degradation leads to the calculation of unpaid restoration costs to meet the injunction of no net degradation of ecosystems. » You could add : « In other words, to estimate an "ecosystem value", we depart from existing monetary valuation which indirectly legitimize a right to exploit ecosystems, to biophysical valuation, which tends to consider the degradation and thus the associated biophysical debt instead.”

Material and methods

To avoid confusion/open-endedness, replace “(XXX,YYY,…)” by “(e.g. XXX or YYY).

Acronyms: Please keep the use of acronyms to a minimum. For instance, DLCT could be used in figure legends only, and the full name should be repeated in the main text (it only appears 11 times in the document). Same for GDP (only used 3 times), ECU (used 10 times), CLC (used 7 times), ENCAT (7 times), or UZHYD (5 times).

Shouldn’t the definition of “sustainable use” be “accessible resources / used resources”? and this should rather be an “index of sustainable use” (between 0 and 1). In fact, shouldn’t it be “accessible resources at t0 / used resources over the studied period”? Even more, if the index is between 0 and 1, shouldn’t it be “used resources over the studied period / accessible resources at t0”?

I also wonder how the authors distinguish accessible and used resources, because the amount of accessible resources will also depend on the “natural” degradation of the resource and its regeneration. In other words, “Accessible resources” could be defined as the sum of “stock – degradation + regeneration + transport (in or out)”, while “used resources” would be defined as the sum of “extraction + storage”, i.e. used resources for human purposes and leaving the Earth cycles, in the short term. But the authors might have another view. Please clarify.

It’s still not clear to me how the health index is calculated.

It seems that similar variables have different names. For instance, in this paragraph (line 246): “Computing synthetic indices of Intensity of Use and Health for each thematic account. Intensity of Use is the ratio of accessible resource to resource use, expressed on a 0 -1 scale. Values higher than 1 mean no resource depletion due to use and are given the value 1. Values between 0 and 1 correspond to situations where use exceeds renewal. The Health diagnosis is based on intermediate indices for water quality, biodiversity change, and other vulnerability factors. ». “ Intensity of Use” looks a lot like the “index of sustainable use” to me. “Health diagnosis” also could be similar to the “health index”. This needs to be clarified (hence the suggestion to used Greek alphabet, if needed).

Results

“2. Ecosystem Water and River Accounts » Please repeat in the text that the analysis applies to a 12-year period so that each subsection is somewhat autonomous. For instance, this sentence “The precipitation regimes have been relatively stable» is wrong unless you add “over the studied period”.

Line 357: “Visual comparison of impacted areas and land cover clearly shows for example that vineyards north of Lyon are concerned, well known for important use of pesticides. » If it’s well known, there should be a scientific reference to quote here. The same applies to the following sentence “It contrasts with small rivers in mountain areas where rivers ecological state has improved during the same period. » Please check carefully that all statements like those are supported by scientific evidence.

Line 380 “The percentage of bio-carbon use with respect to the accessible resource has been estimated at 30-40%.” I think the authors mean that “the index of bio-carbon use” is around 30 to 40% over the studied period. To be more didactic, the authors could add a sentence like “In other words, this means that 30 to 40% of the accessible bio-carbon is extracted from ecosystems for human use or degraded (e.g. deforestation through land use change)”, unless I misunderstood something.

Discussion

Line 534 Please remove this paragraph:

« This work’s three main implications are discussed below:

- The ENCA proof-of-concept – strengths and current limitations,

- Reframing data resources, and

- ENCA and society.”

Because this appear in a subsection of the discussion, and thus at this point, we should only see a suite of paragraphs. No need to explain what each paragraph will deal with beforehand.

The discussion is long and I’m not sure readers will take the required time to read it. There are many interesting items, so it would gain a lot at being more concise.

Tables and figures: As for discussion, consider making the tables and figure legends more concise. Otherwise, readers won’t be read.

Table 2: typo “accesible” (please run a Word error check on the full document)

In tables, could you fuse cells with the same word (e.g. Water should be one single cell). That would make the table less heavy.

Table 3 could be removed (only mention the main numbers in the main text).

Figure 1: Top text “Land cover (ha) or river extent (km)”

Instead of having 3 times "accessible resources" and "resource use", why don’t you put your definition of the index of sustainable use here (your key equation), leading to ecosystem capability.

Letters a and b should be on the left side

There should be a letter c for the map

You should move figure 2a to figure 1 (basic description of the territory)

Figure 2 is not self-explanatory. Here are suggestions to improve:

If11, if12, etc. have not been defined, please remove

Figure 2c: “broad classes” should be replaced by “land cover evolution over the 12-year period”

Figure 2d “drivers of change” should be bold like “land cover evolution over the 12-year period” in 2c

The graphical abstract could be the last figure, and referred to in the discussion.

Left panel: I would be more controversial: “The GDP only considers economic exchanges between humans, and ignores the consumption of natural capital”

“This consumption could be measured trough the changes of Ecological value (biophysical). The ecological value consist of four core accounts.” Could be replaced by “This consumption could be measured through the changes of Ecological, biophysical, value, i.e. water + land cover + bio-carbon + ecosystem infrastructures”

Right side title: “Changes of Ecological value of the Rhône river watershed in 12 years”

Please keep the same color code throughout for “water”, “land cover”, etc.

I don’t understand why the arrows showing “use” are pointing up. Shouldn’t they be pointing down (it’s a loss of capital)?

---

## [Reviewer Report]

*Comments to Author*: Thank you for your resubmission. Please address the minor issues brought forward by the reviewers. I do not think that you need to address reviewer #2's criticism about the fit to the journal. Focus on the suggestions to improve the clarity of the text and figures.

---

## [Reviewer Report]

*Comments to Author*: The reviewers have addressed my remaining concerns. I only have a few minor edits, but I won’t need to re-review the article.

The title is fragmented. How about: “Ecosystem Natural Capital Accounting: A proof-of-concept at a territorial watershed scale” (also because “watershed” alone has other meanings in image analysis and may be confusing for the community)

Small corrections in the graphical abstract:

- a yellow arrow with “artificializa” instead of “artificialization”,

- the text should not be in all directions (in the current version, we need to turn around the graph to read everything)

- “ecosystem infrastructure” with “potential from 0.5 to 30%” decrease is too cryptic to be understandable in such an abstract. Why not more simply “ecosystem services” instead of “ecosystem infrastructures” in this abstract (easier to grasp)?

Article type: The citizen science umbrella might not be relevant here: this method builds on local data, but remains top-down. Since it is a proof of concept, that also calls for more formalization in the future, it might fit better under “theory” format in the end?

There are still some single sentences that should be grouped with the neighboring paragraph to ease the reading.

Figure 1a: some extra corrections (important ones, though):

- To avoid confusion in fig 1a: “Land cover (ha) and river extent (km)”

- The color code and repetitions on figure 1a are not self-explanatory. I would remove all colors, keep the first two lines (land cover…, Ecosystem carbon…) and then below write: “poor indicators for resource accessibility” (only once), then below “questionable indicators for intensity of use and health index” (only once), then below “poor indicators of ecosystem capability”. This would also avoid the confusion with the different color codes in the other figure panels.

Figure 1c: Why not write “Socioecological Landscape Units” (SELU by DLCTs is too cryptic)

---

## [Reviewer Report]

*Comments to Author*: Dear authors,

Thanks for revising your manuscript. I think it is now ready for publication. I just regret that you did not associate a revision letter to this new round of review to explain some of your choices. It seems that you may disagree with some of my comments and it would have been nice to know why. However, it was minor comments and they should not impact the take-home message of your paper. I may miss your letter, in that case, I apologize and you can obviously forget my previous comment.

Best regards,